# Existence of multiple critical cooling rates which generate different types of monolithic metallic glass

Jürgen E.K. Schawe[1] & Jörg F. Löffler [2]

Via fast differential scanning calorimetry using an Au-based glass as an example, we show that metallic glasses should be classified into two types of amorphous/monolithic glass. The first type, termed self-doped glass (SDG), forms quenched-in nuclei or nucleation precursors upon cooling, whereas in the so-called chemically homogeneous glass (CHG) no quenched-in structures are found. For the Au-based glass investigated, the critical cooling and heating rates for the SDG are 500 K s$^{-1}$ and 20,000 K s$^{-1}$, respectively; for the CHG they are 4000 K s$^{-1}$ and 6000 K s$^{-1}$. The similarity in the critical rates for CHG, so far not reported in literature, and CHG's tendency towards stochastic nucleation underline the novelty of this glass state. Identifying different types of metallic glass, as is possible by advanced chip calorimetry, and comparing them with molecular and polymeric systems may help to elaborate a more generalized glass theory and improve metallic glass processing.

[1] Mettler-Toledo GmbH, Analytical, 8606 Nänikon, Switzerland. [2] Laboratory of Metal Physics and Technology, Department of Materials, ETH Zurich, 8093 Zurich, Switzerland. Correspondence and requests for materials should be addressed to J.F.L. (email: joerg.loeffler@mat.ethz.ch)

The kinetic concept of glass formation was developed by Gustav Tammann[1] 85 years ago. According to his assumption, a glass can be formed by cooling if the curves of the temperature dependency of nucleation and growth are significantly separated. The tendency of a metallic alloy to form a glass is usually characterized by the glass-forming ability (GFA), which corresponds to a critical cooling rate, $\beta_{c,cr}$, at which no crystallization occurs during cooling from the melt[2-6]. After cooling at $\beta_{c,cr}$ to temperatures below the glass transition region the material is expected to form a monolithic glass, i.e., a glass whose atomic structure is completely amorphous. The structure of the glass can be changed by annealing it in the glassy state[7-12] or by varying the cooling rate during glass formation[13-16].

Another kinetic phenomenon is the critical heating rate, $\beta_{h,cr}$, at which a glass does not crystallize upon heating[14,15,17]. It may be expected that $\beta_{h,cr} \gg - \beta_{c,cr}$, because nucleation is much more dominant at deep undercooling, i.e., the maximum of the nucleation rate is at lower temperatures than the maximum of the growth rate[14,15], and the critical size of nuclei may decrease at lower temperatures[15].

The competition between glass formation and nucleation is a general phenomenon and occurs not only in rapidly quenched metallic alloys, but in various classes of metastable materials with different types of bonding. While glass formation is relatively rare in metallic alloys, polymers usually form a semicrystalline structure where the macromolecule is part of the crystalline and amorphous phase. Complete crystallization from the melt is thus basically impossible in such systems, so that glass formation in polymers can be considered a universal phenomenon. In polymeric materials the terms amorphous glass and semicrystalline glass are commonly used and distinguished from one another. For metals, a somewhat different terminology is used and the equivalent terms are monolithic glass and metallic glass composite, respectively.

Since the introduction of non-adiabatic chip calorimetry[18,19] and its commercialization[20,21], calorimetric measurements at defined cooling and heating conditions have been possible using fast differential scanning calorimetry (FDSC). This technique is frequently used to study glass transition phenomena and nanostructure formation in polymers[22-24], molecular glass formers[25], and chalcogenides[26-29]. However, recently FDSC has also been identified by the metals community as a suitable method for studying glass formation, nucleation, and phase transitions in metastable metallic alloys[14,30-36]. In[14], one of the authors of this paper reported on the crystallization kinetics of an Au-based bulk metallic glass ($Au_{49}Ag_{5.5}Pd_{2.3}Cu_{26.9}Si_{16.3}$[37]), and constructed, via isothermal measurements in the millisecond range, complete time–temperature–transformation (TTT) diagrams of crystallization in the undercooled/supercooled liquid range upon cooling and heating.

In this study we use a newly developed Flash DSC2+ instrument (see Methods) which allows us to perform calorimetric experiments at ultrafast cooling rates of 40,000 $Ks^{-1}$ and heating rates greater than 60,000 $Ks^{-1}$. The latter also allows us to up-quench a certain phase, where up-quenching denotes a heating process that is so rapid that no structural changes occur before melting of the previously frozen phase[36].

In this work we performed FDSC investigations on the bulk metallic glass (BMG) $Au_{49}Ag_{5.5}Pd_{2.3}Cu_{26.9}Si_{16.3}$ at ultrafast rates with linear heating and cooling. We illustrate here that in the case of metals, amorphous (monolithic) glasses need to be classified into two categories. We term these self-doped glass (SDG) and chemically homogeneous glass (CHG). We show that quenched-in nuclei or nucleation precursors form in the SDG upon cooling at medium rates, which generates significant differences in the critical cooling and heating rates ($\beta_{c,SDG} \approx 500\ K\,s^{-1} \ll \beta_{h,SDG} \approx$

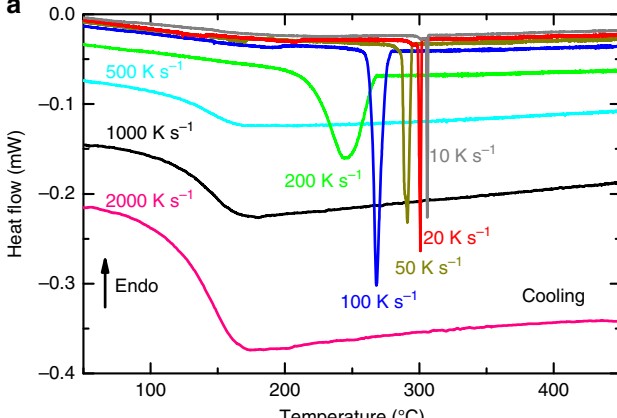

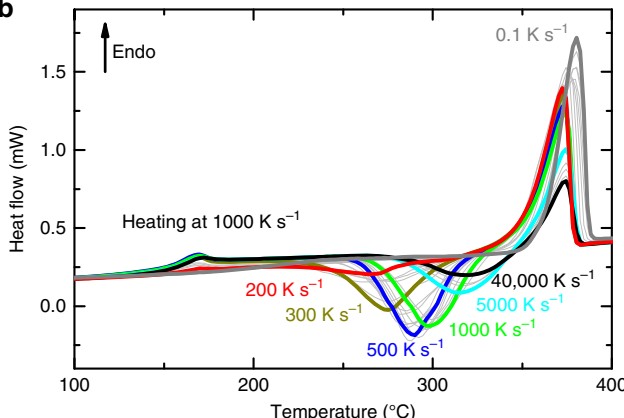

**Fig. 1** Phase transformations upon cooling and heating. **a** Selected cooling curves from the melt measured at various rates. Crystallization occurs at slow cooling up to approximately 200 K s$^{-1}$. At faster cooling the sample forms a glass. **b** Heating curves measured at 1000 K s$^{-1}$ after cooling at various rates between 0.1 K s$^{-1}$ and 40,000 K s$^{-1}$. The previous cooling rates are indicated only for the thickly drawn curves, while curves measured for other cooling rates are thin and gray-colored

20,000 K s$^{-1}$). In contrast, such nuclei no longer form upon rapid cooling, which leads to the fact that the rates of critical cooling ($\beta_{c,CHG} \approx 4000\ K\,s^{-1}$) and critical heating ($\beta_{h,CHG} = 6000\ K\,s^{-1}$) are very similar for the CHG. This similarity has not been reported before because the critical cooling rate measured has always been that of an SDG[38,39]. The ramifications for the understanding of metallic glass processing are, however, significant because the conjectured and/or measured great differences between critical cooling and heating rates have often been explained by a pronounced asymmetry in crystallization behavior[14,15,17]. By analyzing the FDSC results and discussing the commonalities and differences involved in nucleation and glass formation for metallic and polymeric glass formers, we also intend to contribute to the development of a more generalized glass theory.

## Results

**Critical cooling rates and formation of different glasses.** Figure 1a reveals FDSC cooling curves measured at different rates. Crystallization peaks are detected for cooling rates below 500 K s$^{-1}$, and the temperature of the crystallization process increases with decreasing cooling rate. The rate of 500 K s$^{-1}$, where crystallization is no longer evident, is the critical cooling rate, i.e. the minimum cooling rate at which the liquid can form a glass.

Figure 1b shows heating curves at 1000 K s⁻¹ measured after cooling at various rates (between 0.1 and 40,000 K s⁻¹). The glass transition takes place at about 160 °C, and the exothermic crystallization peak accrues in the temperature range between 260 and 315 °C followed by an endothermic melting peak. The crystallization shows a strong dependence on the previous cooling conditions, an effect that has been seen frequently in various materials[14,32,40]. The temperature of the crystallization peak, $T_{Peak}$, the intensity of the glass transition (step height $\Delta c_p$), and the total enthalpy, $\Delta H$, (containing crystallization and melting) are evaluated from these curves.

The total enthalpy $\Delta H$ reflects the crystallinity of the sample after the cooling process[41,42]. In a first approximation the crystallinity is $\alpha = \Delta H / \Delta H_m$, where $\Delta H_m$ is the equilibrium melting enthalpy measured using a completely crystallized material which was previously cooled at 1 K s⁻¹. The actual melting enthalpy measured, $\Delta H_{melt}$, is usually smaller than the equilibrium melting enthalpy, $\Delta H_m$. This is because the phase which eventually melts is not necessarily the stable phase, and/or the material has not completely crystallized before melting. In the case of an amorphous specimen, $\Delta H = \Delta H_{cryst} + \Delta H_{melt}$ is practically zero, i.e., $\alpha = 0$, because the enthalpy of possible exothermic crystallization processes during heating is the same as the enthalpy of melting of these crystals. Small differences are generated by the heat capacity contribution[43]. On the other hand, $\alpha = 1$ for a fully crystallized sample.

The increase in the heat capacity at the glass transition, $\Delta c_p$, is the intensity of the glass transition, which also depends on crystallinity. The relative intensity of the glass transition, $\Delta c_p / \Delta c_{p,a}$, has been found to depend linearly on the crystallinity $\alpha$ (see Supplementary Note 1 and Supplementary Fig. 1), where for the completely amorphous phase the intensity of the glass transition is $\Delta c_{p,a} = 0.14$ J g⁻¹ K⁻¹ (see Fig. 2b). Such a linear dependence is expected for semi-crystalline materials with a single type of amorphous phase and has also been observed for molecular glasses, in contrast to macromolecular glasses[44]. Thus, the relative glass transition intensity, $\Delta c_p / \Delta c_{p,a}$, can be used as a measure for the content of amorphous phase in semi-crystalline glasses (SCGs). This is important because for small crystallinities the change in $\Delta c_p$ is often more sensitive than the variation in transformation enthalpy.

Measurements similar to those in Fig. 1 were recently performed on a Ce-based BMG[30], where the authors evaluated the crystallization and melting events of the heating curves obtained by nanocalorimetry. They interpreted the data by introducing two critical cooling rates, and found that during slow cooling only the crystalline phase occurred. After cooling at medium rates, i.e., at those between the two critical cooling rates identified, they observed a mixed glassy-crystalline structure, while very rapid cooling above the second critical cooling rate suppressed crystallization[30]. In our work, we performed experiments with a sample mass on the order of micrograms to avoid critical size effects[14,45] and also evaluated the glass transition in detail. We avoided potential oxidation effects by investigating a precious-metal-based BMG. By significantly refining the measurements we were thus able to distinguish between two different types of amorphous glasses, as discussed in the following.

Figure 2a plots the temperature of the crystallization peaks observed during heating at a rate of 1000 K s⁻¹ vs. the cooling rate at which the glass was formed (Fig. 1b). At slow cooling rates (<200 K s⁻¹) no crystallization is observed in the heating curves because the sample completely crystallized during cooling. After fast cooling (≥4000 K s⁻¹) the temperature of the crystallization peak is constant and amounts to 314 °C, i.e., in this range the crystallization is invariant with respect to the previous cooling rate or, in other words, the cooling condition required to form a

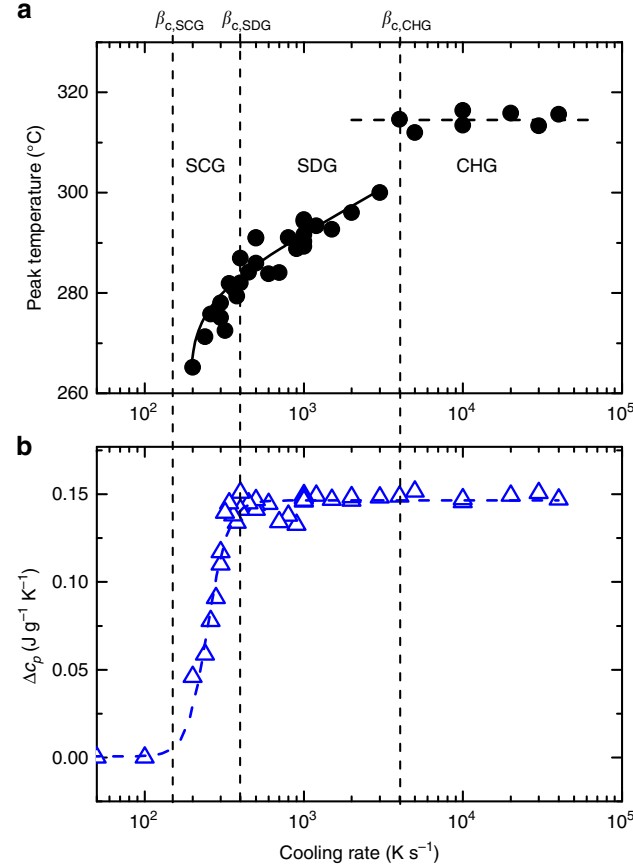

**Fig. 2** Cooling rate dependence of the crystallization peak and glass transition. Data derived from the heating curves measured at 1000 K s⁻¹ vs. cooling rate at which the glass was formed (see Fig. 1b). **a** Peak temperature of crystallization event. **b** Intensity of the glass transition, $\Delta c_p$. The solid curve in **a** results from a fit using Eq. (1); the dashed curves in **a** and **b** are guides for the eye

glass has no influence on the crystallization process during heating. We assume that here no nuclei have formed during cooling, so all nucleation must occur during heating.

After cooling at rates between 200 and 4000 K s⁻¹ the crystallization temperature, $T_{Peak}$, is significantly lower than that after fast cooling, and in this range $T_{Peak}$ increases with increasing cooling rate. In these more slowly cooled glasses, the crystallization is accelerated and may be interpreted by the formation of (nano)structures during cooling such as quenched-in nuclei[46] or precursors of nucleation.

If a glass contains a sufficiently high number of quenched-in nuclei, heterogeneously nucleated crystallization occurs during heating. Based on the Kolmogorov–Johnson–Mehl–Avrami (KJMA) equation[47–49] for isothermal crystallization we can show that the cooling rate dependence of the peak temperature follows a power law (see Supplementary Discussion), i.e.,

$$T_{Peak} = T_1 + C|\beta_c - \beta_0|^\kappa, \qquad (1)$$

where $C$ and $\kappa$ are empirical constants, $T_1$ is the minimum crystallization temperature during heating, $\beta_0$ is the minimum cooling rate for crystallization during heating, and $\beta_c$ is the cooling rate.

By fitting of the data in Fig. 2a, the parameters are determined as $C = 34.2$ K, $\kappa = 0.1$, $T_1 = 224.9$ °C, and $\beta_0 = 190.7$ K s⁻¹. The resulting curve fits the experimental data well. This indicates that the nucleation density is, in a first approximation, proportional to

the previous cooling rate. This agrees with investigations on a Zr-based BMG, where model calculations indicate that the crystallization kinetics during heating can be described by a number density of pre-existing nuclei rather than by a nucleation rate[50].

Figure 2b shows the intensity of the glass transition as a function of the previous cooling rate. Below a cooling rate of approximately 200 K s$^{-1}$ no glass transition occurs, and the material is completely crystalline. After cooling at a rate of more than 500 K s$^{-1}$ the glass transition intensity is constant ($\Delta c_p = 0.14$ J g$^{-1}$ K$^{-1}$) and the sample is completely amorphous. At cooling rates between these limits, crystals have formed but the crystallization time is too short for complete crystallization of the sample. The reaming glassy phase generates a reduced glass transition with $\Delta c_p < 0.14$ J g$^{-1}$ K$^{-1}$. The intensity of the glass transition vs. the previous cooling rate to form a glass in Fig. 2b, and the shift of the crystallization temperature in Fig. 2a, indicate the existence of three critical cooling rates.

The lowest critical cooling rate, $\beta_{c,SCG} \approx 200$ K s$^{-1}$, is the upper cooling rate limit at which the material completely crystallizes during cooling. Above this rate (and below 500 K s$^{-1}$) crystals and glassy regions coexist, and a semi-crystalline glass (SCG) or glass composite (using metallurgy terminology) forms. Such a glass often has the structure of a dispersion of nanoparticles in the amorphous matrix[51].

The second critical cooling rate, $\beta_{c,SDG} \approx 500$ K s$^{-1}$, is the lower cooling rate limit at which an amorphous glass can be formed. Above this rate (and below 4000 K s$^{-1}$), the glass contains regions with an increased local order, which comprises quenched-in nuclei or precursors for nucleation (embryos). These metastable structures accelerate nucleation or act directly as nuclei during subsequent heating. We call such a glass self-doped glass (SDG). The mass content of the quenched-in nuclei or precursors is so small that it has no influence on the glass transition intensity (Fig. 2b). The increase in the crystallization temperature with increasing cooling rate, however, indicates a reduced number of quenched-in nuclei or precursors after faster cooling (Fig. 2a).

The SDG contains nanostructures which may be non-crystalline metastable clusters in the glass resulting from spatial heterogeneities[11,52,53] or quenched-in nuclei, as frequently noted in Al-based BMGs[54–59]. Such nuclei form during quenching, but their growth is limited due to the reduced temperature and solute rejection[55]. Transmission electron microscopy (TEM) and X-ray diffraction (XRD) measurements have shown that a BMG appears completely amorphous even if it contains quenched-in nuclei[56]. The number density of such structures in an Al-based BMG is on the order of $10^{21}$ m$^{-3}$ (ref. [60]).

Above the highest critical cooling rate of $\beta_{c,CHG} \approx 4000$ K s$^{-1}$, a chemically homogeneous glass (CHG) forms. Such a glass contains no quenched-in nuclei or precursors which accelerate nucleation during heating.

A comparison of the crystallization kinetics of the SCG and the SDG in Fig. 2a shows no abrupt variations in the cooling-rate dependence. In fact, for both glasses the cooling rate dependence of the crystallization peak follows Eq. (1) with the same parameter set, i.e., the nucleation kinetics in the SCG and the SDG are comparable. This indicates that the crystals and the quenched-in clusters act similarly in the crystallization process, which leads us to conclude that the quenched-in clusters are nuclei rather than precursors of nucleation at low temperatures.

In contrast to the above, a comparison of the crystallization kinetics of SDG and CHG in Fig. 2a clearly shows a cooling-rate dependence of the crystallization peak during heating for these two types of amorphous glasses. Whereas in the CHG crystallization is independent of cooling-rate variation, in the SDG a lower previous cooling rate results in accelerated crystallization. The influence of processing and cooling conditions on

crystallization of glasses during heating has been reported for many different families of BMGs[14,40,50,61,62]. This leads us to conclude that the formation of SDGs is a common occurrence in BMG alloy processing.

To study the generality of this phenomenon, we also performed similar measurements on the Pt-based BMG Pt$_{57.3}$Cu$_{14.6}$-Ni$_{5.3}$P$_{22.8}$, as described in Supplementary Note 2. Here we can also distinguish between SCG, SDG, and CHG and obtain critical cooling rates of $\beta_{c,SCG} \approx 2$ K s$^{-1}$, $\beta_{c,SDG} \approx 20$ K s$^{-1}$, and $\beta_{c,CHG} \approx 150$ K s$^{-1}$, respectively, as seen in Supplementary Fig. 2. The occurrence of SDG and CHG thus appears to be a general phenomenon in BMG processing.

**Critical heating rates.** If a glass is heated at sufficiently high rates, no crystallization occurs. The critical heating rate is the lower heating rate limit for avoiding crystallization, and it has been frequently reported that it is substantially higher (in orders of magnitude) than the critical cooling rate[14,15,45,63,64]. The existence of differently structured glasses (SDG and CHG), however, leads us to expect that each type of glass has its characteristic critical heating rate.

To determine the critical heating rate of the CHG, the Au-based liquid was cooled at 20,000 K s$^{-1}$ and the resulting glass reheated at various rates. Selected curves are shown in Fig. 3a. At low heating rates crystallization occurs after the glass transition, followed by melting. The crystallization peak shifts to higher

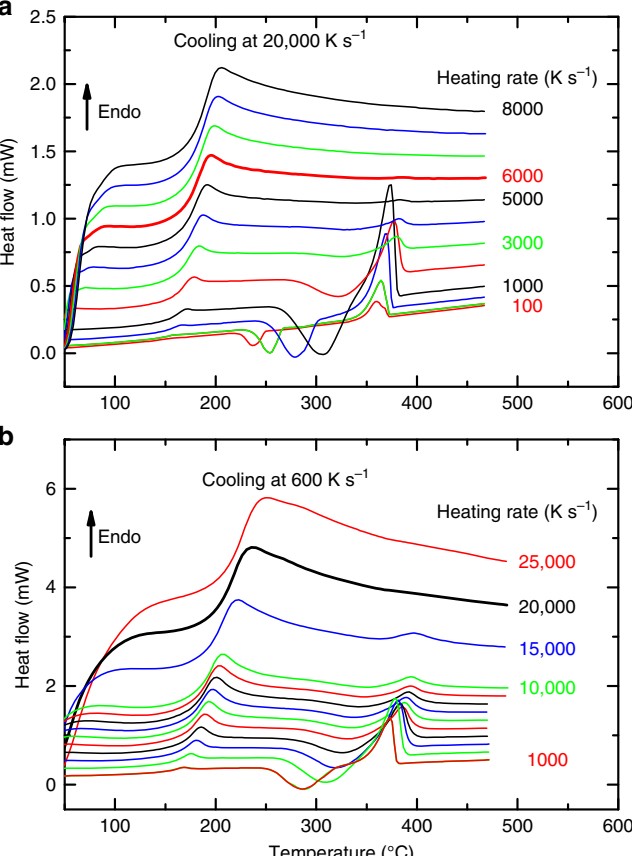

**Fig. 3** Heating behavior of CHG and SDG. Selected heating curves measured at various rates, as indicated in the figure. **a** Chemically homogeneous glass (CHG) that was formed at a cooling rate of 20,000 K s$^{-1}$. **b** Self-doped glass (SDG) that was formed at a cooling rate of 600 K s$^{-1}$. The curves measured at the critical heating rates (6000 K s$^{-1}$ for CHG in **a** and 20,000 K s$^{-1}$ for SDG in **b**) are illustrated as bold curves

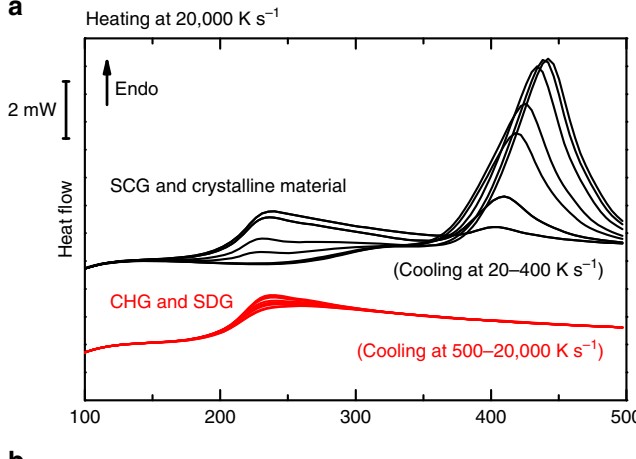

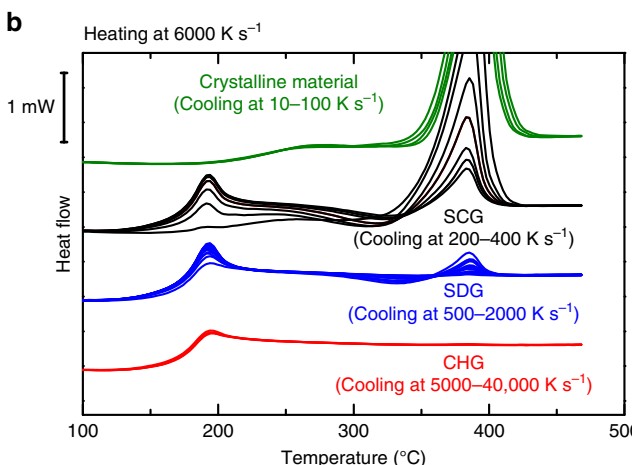

**Fig. 4** Separation of different types of glasses. **a** Heating curves measured at 20,000 K s$^{-1}$ for differently cooled samples between 20 and 400 K s$^{-1}$ (black) and 500 and 20,000 K s$^{-1}$ (red). **b** Heating curves measured at 6000 K s$^{-1}$ for differently cooled samples between 10 and 40,000 K s$^{-1}$. This measurement procedure enables us to distinguish four different classes of materials according to the crystallization and melting events

temperatures when the heating rate is increased, and at a heating rate between 1000 and 5000 K s$^{-1}$ crystallization is not finished when melting starts. The material cannot crystallize completely, i.e., the crystallization rate is thermodynamically limited. Nevertheless, for all crystallization and melting processes the total transformation enthalpy (containing crystallization and melting) becomes zero within the error of the experimental detection limit. Above the critical heating rate of $\beta_{h,CHG} \approx 6000$ K s$^{-1}$ no crystallization can be observed. In that case, also no melting event occurs[14].

To determine the critical heating rate, $\beta_{h,SDG}$, of an SDG the liquid was cooled at 600 K s$^{-1}$ to RT before heating the glass at various rates (Fig. 3b). The curves appear similar to those in Fig. 3a, but the effect of neither crystallization nor melting appears at a much higher critical heating rate of $\beta_{h,SDG} \approx 20,000$ K s$^{-1}$.

The assumption that in metallic glasses the critical heating rate is orders of magnitude higher than the critical cooling rate[15,63,64] is thus only valid for the nanostructured SDG ($\beta_{c,SDG} \approx 500$ K s$^{-1}$, $\beta_{h,SDG} \approx 20,000$ K s$^{-1}$), where quenched-in nuclei or precursors already exist. Here, to completely melt the glassy phase the heating process has to be fast enough so that existing nuclei stay at subcritical size. This happens when the heating rate is so high that the increase in the critical size of a nucleus due to temperature

increase is faster than the actual nucleation, i.e.,

$$-\frac{\beta_h}{V}\frac{dv^\star}{dT} > \frac{dN}{dt}, \tag{2}$$

where $\beta_h$ is the heating rate, $V$ is the volume, $v^\star$ is the volume of a critical nucleus, and $N$ is the number of nuclei. The second possible mechanism for avoiding crystallization of a glass with existing nuclei is to bypass the growth region during heating. In the framework of a TTT diagram this means that the curve of the total temperature program does not touch the growth region.

For CHGs the critical cooling and heating rates only differ by a factor of 1.5 ($\beta_{c,CHG} \approx 4000$ K s$^{-1}$, $\beta_{h,CHG} \approx 6000$ K s$^{-1}$). This relatively small difference can be explained by an asymmetric curve shape of the nucleation rate vs. temperature, or by a very small number of remaining quenched-in nuclei or precursors.

In the next step the crystallization and melting behavior of samples cooled at different rates between 10 and 40,0000 K s$^{-1}$ was investigated upon heating at the two critical heating rates of $\beta_{h,SDG} = 20,000$ K s$^{-1}$ and $\beta_{h,CHG} = 6000$ K s$^{-1}$ (Fig. 4). The heating curves at 20,000 K s$^{-1}$ (Fig. 4a) are arranged in two groups. After fast cooling ($\beta_c \geq \beta_{c,SDG} \approx 500$ K s$^{-1}$) no crystallization occurs and only the glass transition is measured. For slow cooling at rates $\beta_c < \beta_{c,SDG}$ the formed glass contains crystals, which reduces the intensity of the glass transition and causes a melting peak. No crystallization is observed here because of the high heating rate of 20,000 K s$^{-1}$, which is fast enough to prevent not only the crystallization of quenched-in structures, but also the further growth of existing crystals. At this high heating rate it is only possible to distinguish between amorphous glasses (CHG and SDG) on the one hand and semicrystalline glasses (SCGs) or fully crystalline material on the other. This information can also be derived from the cooling curves in Fig. 1a and the intensity of the glass transition in Fig. 2b.

Figure 4b shows heating curves of differently cooled samples measured at a heating rate of 6000 K s$^{-1}$. Here the curves can be separated into four groups.

In the first group, where $\beta_c \geq \beta_{c,CHG} \approx 4000$ K s$^{-1}$, no crystallization and melting events occur in the heating curve. The heating rate is fast enough to up-quench the chemically homogeneous glass. Here, the term up-quenching refers to a heating process at which the heating is so rapid that no structural changes occur before melting of the previously frozen phase[36].

At $\beta_{c,SDG} \approx 500$ K s$^{-1} \leq \beta_c < \beta_{c,CHG}$, the heat flow around the glass transition region is independent of the previous cooling rate. This means that the glass transition intensity does not vary with the cooling rate (although changes are seen in the glass transition region due to a variation in the enthalpy recovery peak, resulting from the different cooling histories of the SDG). Furthermore, crystallization and melting events can be detected in the supercooled melt above the glass transition, but the total transition enthalpy remains zero. The quenched-in clusters act during heating as nuclei or precursors for nucleation, which enables crystallization.

For $\beta_{c,SCG} \approx 200$ K s$^{-1} \leq \beta_c < \beta_{c,SDG}$, the glass transition intensity reduces with decreased cooling rate. Here, crystallization also occurs during heating but the melting peak is significantly higher than the crystallization peak because the SCG already contains crystals which melt, but no longer contribute to the crystallization event upon heating.

Finally, after slow cooling at $\beta_c < \beta_{c,SCG}$, no glass transition and crystallization are observed upon heating because the material already crystallized completely during cooling. A slight decrease in the melting peak temperature at faster cooling rates reveals the reduced stability of the crystals formed. This indicates that the

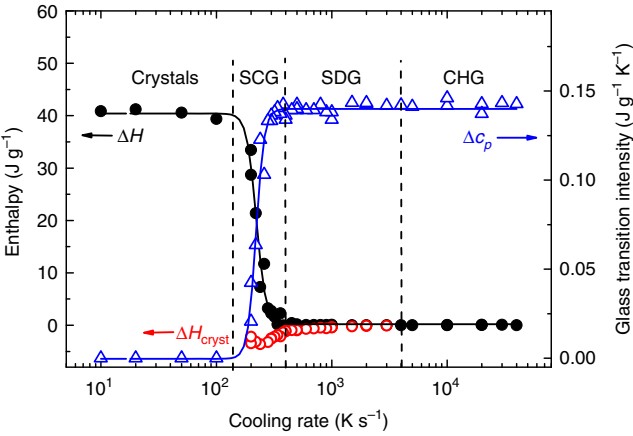

**Fig. 5** Indication of different structures. Total enthalpy of the phase transformation ($\Delta H$), enthalpy of crystallization during heating ($\Delta H_{cryst}$) and intensity of the glass transition ($\Delta c_p$) upon heating at 6000 K s$^{-1}$ (Fig. 4b) vs. cooling rate at which the glass was formed. Four different states can be clearly observed: the fully crystalline region, the semi-crystalline glass (SCG), the self-doped glass (SDG), and the chemically homogeneous glass (CHG)

crystals have sizes in the nanometer range, and that the melting temperature of the nanocrystals decreases with a decrease in particle size due to their surface enthalpy (Gibbs–Thomson effect). A melting point depression has, for example, been experimentally verified for nanosized indium particles[65].

The intensity of the glass transition, $\Delta c_p$, the total transformation enthalpy, $\Delta H$, and the enthalpy of the exothermic crystallization peak, $\Delta H_{cryst}$, were evaluated from the curves in Fig. 4b. The 6000 K s$^{-1}$ heating rate data regarding $\Delta c_p$ and $\Delta H$ were also added to Supplementary Fig. 1 to further verify the linear correlation between the relative intensity of the glass transition and the crystallinity. Figure 5 shows all data ($\Delta H$, $\Delta H_{cryst}$, $\Delta c_p$) as a function of the cooling rate at which the glass was formed, and clearly links the four different regimes with the cooling process. In the CHG no crystallization and melting occur and the intensity of the glass transition is maximal. The latter is also true for the SDG, but here a small crystallization event is observed, and the modulus of the crystallization enthalpy $|\Delta H_{cryst}|$ slightly increases with a decrease in cooling rate. However, because the total enthalpy of transformation is zero, all crystals must have formed upon heating (whereas no crystals had formed upon cooling). The crystallization process is accelerated (see Fig. 2a) due to the existence of quenched-in nuclei, which, however, do not influence the intensity of the glass transition.

In the SCG the crystallinity increases with a decrease in cooling rate, which is reflected in the related increase in the total transformation enthalpy and the decrease in the glass transition intensity. During heating the remaining amorphous fraction tends to crystallize, which generates a finite crystallization enthalpy. Finally, in the crystalline material created after slow cooling, only melting can be measured in the subsequent heating curve, so that $\Delta c_p = 0$ and $\Delta H = \Delta H_{melt} \approx \Delta H_m$ reaches a maximum.

These results agree nicely with the findings in Fig. 2, so that recommendations regarding the experimental parameters for a detailed analysis of glass behavior can be derived: In cooling experiments it is only possible to determine the critical cooling rate for SDGs (see Fig. 2b), while more information on glass properties can only be obtained from heating measurements. Here, the heating rate selected is essential for the resolution of the measurement, and the rate must be adapted to the kinetics of the

nucleation and growth processes. If the heating rate is $\geq\beta_{h,SDG} = 20{,}000$ K s$^{-1}$ (Fig. 4a), the results will only duplicate the conclusions of the cooling measurements. If the heating rate is too slow, nucleation will always occur during heating and the resolution in the kinetic analysis will not be sufficient to differentiate the various initial conditions in the glass. At heating rates $\beta_{h,CHG} \leq \beta_h < \beta_{h,SDG}$ (Fig. 4b), the enthalpy determination by peak integration (see Fig. 5) can be used. This is more sensitive and robust than the evaluation of the peak temperature (Fig. 2a), because it is an integral method without any curve shape influence.

**Isothermal crystallization**. The non-isothermal measurements indicate that two different kinds of completely amorphous glass can be formed during cooling. The different nucleation densities in the glasses also influence the isothermal crystallization kinetics. To demonstrate this, CHG and SDG were formed upon cooling at 20,000 K s$^{-1}$ and 500 K s$^{-1}$, respectively, and heated at a rate of 30,000 K s$^{-1}$ to a crystallization temperature between 179 and 349 °C (see Supplementary Note 3). The heat-flow curves of isothermal crystallization measured within a timeframe of 5 s are shown in Supplementary Fig. 3. Most curves show multiple crystallization events, which are possibly induced by the formation of different phases. Multiple peaks during isothermal crystallization are frequently reported for BMGs[51,66]. Here we concentrate on the first event.

In the isothermal crystallization curves of the CHG (Supplementary Fig. 3a) the shape of the crystallization peaks and their positions fluctuate significantly. This can be explained by a lack of nuclei in the supercooled melt, with the heat-flow curves reflecting the statistics of spontaneous nuclei formation. This causes the stochastic and non-predictable appearance of the crystallization peaks seen in Supplementary Fig. 3a.

In contrast to the above, the crystallization curves of the SDG (Supplementary Fig. 3b) show smoothed crystallization peaks, as expected for nucleated material or materials with a high nucleation rate. This means that the glass already contains (quenched-in) nuclei, or at least precursors which require only a small amount of free enthalpy to reach the critical size.

**Time–temperature–transformation diagram**. To characterize the crystallization process, from the heat-flow curves in Supplementary Fig. 3 we evaluated the onset times (beginning of crystallization) and peak times of the crystallization (maximum crystallization rate) as a function of temperature. The endset was not selected due to the potential influences of subsequent crystallization processes. Using these data, we can construct time–temperature–transformation (TTT) diagrams (see Fig. 6) for the SDG, obtained by cooling at a rate of 500 K s$^{-1}$, and the CHG, obtained by cooling at a rate of 20,000 K s$^{-1}$. While the TTT diagrams of the SDG and CHG are close at low temperatures, which is the region of high nucleation rate and growth-controlled crystallization, they are significantly different at medium and high temperatures, where the nucleation rate is low. There the SDG crystallizes significantly faster than the CHG. Furthermore, crystallization of the CHG becomes stochastic near and above the nose of the TTT diagram (see region marked in yellow) as the result of a stochastic nucleation process, and at high temperatures the supercooled melt from the CHG does not even crystallize in the timeframe of the investigation (see region marked in light blue) due to a lack of nuclei. A TTT diagram on heating was also measured in ref. [14] using a comparable Au-based glass cooled at a rate of ≤5000 K s$^{-1}$. There, however, the TTT diagram of an SDG was determined without detecting stochastic nucleation.

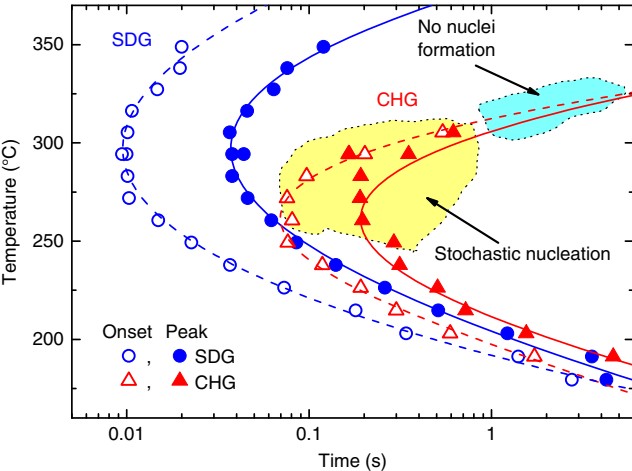

**Fig. 6** Time–Temperature–Transformation diagram. TTT diagram of crystallization after heating from the glassy phase at a rate of 30,000 K s$^{-1}$. The glasses SDG and CHG were formed after cooling at two different rates: 500 K s$^{-1}$ (to generate an SDG) and 20,000 K s$^{-1}$ (to generate a CHG). Unfilled symbols represent the onset times, and filled symbols represent the peak times of crystallization. The region marked in yellow illustrates stochastic nucleation of the CHG, and that in light blue illustrates the complete lack of nuclei formation in the timeframe of 5 s isothermal holding (see Supplementary Fig. 3)

**Visualization of the critical scanning rates for the different types of glasses**. If the melt of a BMG alloy is quenched into the glassy state, nanostructured heterogeneities may form in the glass and influence the glass properties and crystallization kinetics above the glass transition. These are metastable (nano)structures, such as nanocrystals, nuclei, or precursors for nucleation, which significantly influence the crystallization during reheating from the glassy state. The nature of these heterogeneities is different to that of dynamic heterogeneities discussed in the context of cooperative motions in supercooled liquids[67].

Based on TTT diagrams, Fig. 7 provides a schematic visualization of the cooling conditions (solid lines) which generate the different types of glasses. This figure also shows the heating conditions needed to avoid crystallization (dashed lines). The start and end temperatures of the heating and cooling lines in Fig. 7 are always at a temperature slightly above the equilibrium melting temperature and at a temperature in the glassy state, respectively. In this model we assume a thermodynamically equilibrated homogeneous (unstructured) melt above the melting point. This means that the cooling process starts without any heterogeneous nuclei. We also assume a sufficiently large temperature distance between the maxima in growth and nucleation rate.

Upon slow cooling, the melt completely crystallizes and no glass is formed. Upon an increase in the cooling rate, a critical rate, $\beta_{c,SCG}$, is reached, which is the minimum cooling rate at which the material does not completely crystallize. Such an SCG contains both crystalline and amorphous regions. The primary crystallization occurs in the range between the onset and end of the growth curve, but not above the nucleation curve. Figure 7a demonstrates that the critical cooling rate $\beta_{c,SCG}$ is obtained from the intersection of the endset of the growth curve with the (onset of the) nucleation curve. The high-temperature limit for crystallization is the nucleation curve and the low-temperature limit is the curve of growth onset. The shorter the time the material stays in this region, the less is the crystallinity. The heating rate needed to up-quench an SCG (i.e., to melt it without further crystallization) must bypass the growth region; this heating rate is

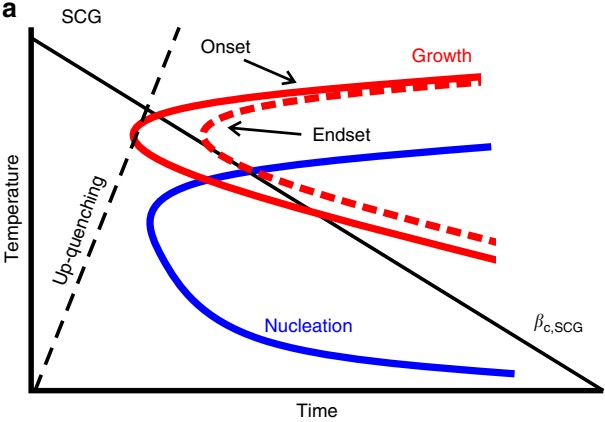

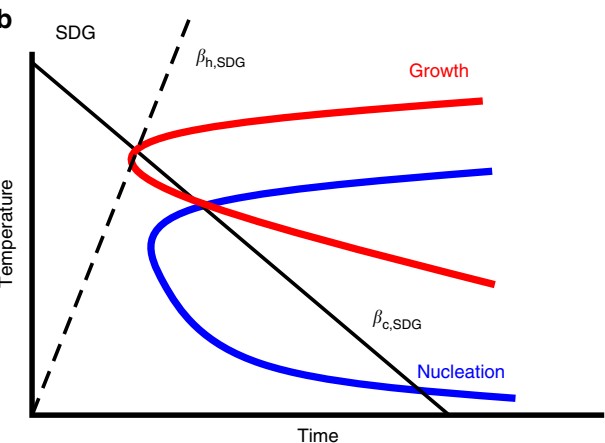

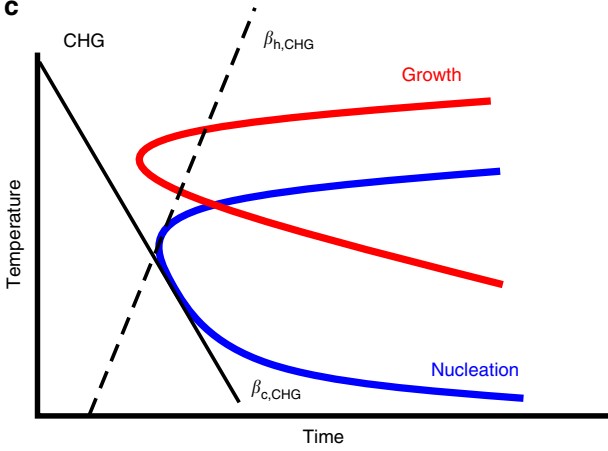

**Fig. 7** Visualization of the critical scanning rates. Schema of the critical cooling rates (solid lines) and critical heating rates (dashed lines) in a TTT diagram. The curves represent the characteristics of growth and nucleation. **a** Critical cooling rate of a semi-crystalline glass (SCG); the growth behavior is characterized by the onset (solid) and the endset (dashed) curves. Also illustrated is the strategy of up-quenching to melt a previously frozen (semi-)crystalline state without any further structural changes upon rapid heating. **b** Critical cooling and heating rates for forming a self-doped glass (SDG). **c** Critical cooling and heating rates of a chemically homogeneous glass (CHG)

illustrated as dashed line in Fig. 7a. Such a strategy of up-quenching a previously frozen metastable structure with the help of advanced chip calorimetry may lead to the discovery of various metastable phases (including their melting enthalpy and temperature)[36], and thus contribute to the understanding of

crystallization pathways not only in metallic systems, but also in polymers, pharmaceuticals and biological systems.

Figure 7b shows the critical conditions for an SDG. The critical cooling rate, $\beta_{c,SDG}$, at which the melt no longer crystallizes is characterized by the intersection point of the growth and nucleation curve. At such cooling the material remains in the nucleation region for a certain time and, consequently, nuclei are formed during cooling. The number of (quenched-in) nuclei in such SDGs depends on the time during cooling in the nucleation region. The critical heating rate, $\beta_{h,SDG}$, has to bypass the growth region. It is the same as the up-quench rate for generating an SCG, and is significantly greater than the critical cooling rate $\beta_{c,SDG}$.

Figure 7c illustrates the critical cooling and heating conditions for a CHG. At the critical rates $\beta_{c,CHG}$ and $\beta_{h,CHG}$ the nucleation region is bypassed and, thus, quenched-in nuclei do not form in CHGs. Because of the lack of nuclei, the material can pass through the growth region during heating without any crystallization. To consider the dwell time of the supercooled liquid in the temperature range of the maximum nucleation rate, the heating line in Fig. 7c does not start at the origin of the coordinates. Consequently, the critical heating and cooling rates for a CHG differ somewhat (in our case by a factor of 1.5), but not by orders of magnitude as reported frequently in literature by determining the thermodynamics of an SDG[14,15,63,64].

While scanning electron microscopy, X-ray diffraction or conventional high-resolution TEM can detect metastable microstructures and nano-scale precipitates in an SCG[68], these techniques may not be able to resolve structural inhomogeneities in an SDG. Here electron correlation microscopy, as recently applied to BMGs[69], may resolve the nuclei in SDGs that clearly influence the crystallization kinetics, as observed by FDSC. Revealing atomic/nano-scale differences between SDGs and CHGs via imaging techniques may be the focus of future work.

## Discussion

An $Au_{49}Ag_{5.5}Pd_{2.3}Cu_{26.9}Si_{16.3}$ metallic glass quenched from the melt into the glassy state under controlled linear cooling was measured over a wide range of cooling rates by fast differential scanning calorimetry (FDSC). The critical cooling rate at which the sample does not crystallize can be directly determined from the cooling curves measured. More information, however, was obtained by analyzing the subsequent heating scans. This study shows that different kinds of glass can be formed via cooling from the melt, and that three different critical cooling rates for metallic glass formation exist: $\beta_{c,SCG} \approx 200 \, \text{K s}^{-1}$, $\beta_{c,SDG} \approx 500 \, \text{K s}^{-1}$, and $\beta_{c,CHG} \approx 4000 \, \text{K s}^{-1}$. The crystallization behavior of these glasses diverges completely.

Cooling between $\beta_{c,SCG}$ and $\beta_{c,SDG}$ produces a semi-crystalline glass (SCG) or glass composite, respectively. The crystalline phase content increases with a decrease in the cooling rate. This is measured using FDSC by the change in the intensity of the glass transition, $\Delta c_p$, or in the total enthalpy of phase transformation, $\Delta H$.

A self-doped glass (SDG) is formed at cooling rates between $\beta_{c,SDG}$ and $\beta_{c,CHG}$. Such a glass appears completely amorphous, but it contains nanoclusters which act as quenched-in nuclei or precursors (embryos) and accelerate crystallization during reheating, even though the glass transition intensity is maximal. Here, the critical heating rate $\beta_{h,SDG} = 20{,}000 \, \text{K s}^{-1}$ (which is the minimum rate to melt the SDG without crystallization) is 40 times greater than the critical cooling rate $\beta_{c,SDG}$. It seems that most of the technically cooled bulk metallic glasses are SDGs, and that the critical cooling and heating rates reported so far always relate to SDGs[14,38,39,61].

Cooling at rates higher than $\beta_{c,CHG}$ produces a chemically homogeneous glass (CHG) without such quenched-in nuclei, and

thus crystallization ability is significantly reduced. For such a CHG the critical heating rate $\beta_{h,CHG} = 6000 \, \text{K s}^{-1}$ is much lower than that of an SDG, and is comparable to the critical cooling rate $\beta_{c,CHG}$.

In conclusion, we emphasize that such detailed distinctions between different types of glasses upon cooling have never been made so far. In fact, they are only now possible because of newly developed chip calorimetry which allows cooling rate variations of up to $40{,}000 \, \text{K s}^{-1}$. The occurrence of SDG and CHG appears to be a general phenomenon in metallic glass formation, as also revealed by additional measurements on a Pt-based BMG (see Supplementary Figure 2). The distinction between SDGs and CHGs has important consequences for metallic glass processing, and for glass theory in particular, in view of the fact that many publications have reported on various glass-forming criteria without drawing on detailed experimental knowledge of the glass nanostructure or medium-range order. The concept of differently structured glasses presented here is based on fundamental models of glass formation and nucleation, and may also hold for other kinds of thermal treatment (e.g., short annealing above the glass transition). We therefore assume that SCG, SDG and CHG can also form in molecular glasses in dependence on their preparation condition. We thus also expect this phenomenon to be highly relevant to the physical stability of molecular glasses, which are used in many areas such as pharmaceuticals and biology.

## Methods

**Sample preparation**. The elements Au (99.95%), Cu (99.9%), Ag (99.5%), Si (99.95%), and Pd (99.95%) were pre-alloyed to a sample with nominal composition $Au_{49}Ag_{5.5}Pd_{2.3}Cu_{26.9}Si_{16.3}$ (in at.%) via repeated induction melting in a quartz tube that was sealed under 99.999% pure Ar atmosphere[14]. The mass loss during alloying was negligible, so that the real composition equaled the nominal one. Chemically homogenous glassy ribbons of approximately 30 μm thickness were then produced by melt spinning under 5N+ pure He atmosphere.

**Fast differential scanning calorimetry**. FDSC measurements were performed using a prototype of a Mettler-Toledo Flash DSC 2+. This instrument is an advanced version based upon Flash DSC technology, with a sealed measurement cell for reduced oxygen content. The Flash DSC 2+ can be operated with a UFS 1 sensor[70] or a new UFH 1 sensor (see insert to Supplementary Fig. 4), which can be operated to about 1000 °C. The active area of the latter sensor is reduced by 1/3 and the membrane thickness is reduced compared to the UFS 1 sensor. The diameter of the active zone is approximately 100 μm, which increases the applicable cooling rate to approximately $40{,}000 \, \text{K s}^{-1}$.

The sample support temperature of the FDSC was set to −30 °C using a Huber intracooler TC45, and the furnace was purged with nitrogen at a flow rate of 60 ml min⁻¹. The melt-spun ribbons were cut under a stereomicroscope into small pieces with a surface of approximately $10^4 \, \mu m^2$. Their mass was estimated using the melting enthalpy of $\Delta H_m = 40.4 \, \text{J g}^{-1}$ (ref. [14]) to be on the order of 1 μg, for which no size-dependent nucleation and crystallization effects are expected[45]. Details regarding temperature calibration, thermal lag (Supplementary Fig. 4), and enthalpy resolution (Supplementary Fig. 5) are given in the Supplementary Methods section.

## Data availability

The datasets generated during and/or analysed during the current study are available from the corresponding author on reasonable request.

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

## Acknowledgements

This work was created because a person in industry and another in academia shared the same enthusiasm for science. We thank ETH Zurich and Mettler-Toledo for supporting this type of collaboration without the need of external funding.

## Author contributions

J.E.K.S and J.F.L. conceived the study. J.F.L. provided the material and J.E.K.S conducted the measurements. Both authors evaluated and discussed the data extensively and jointly wrote the manuscript.

## Additional information

**Competing interests:** The authors declare no competing interests.

