## [Peer Review File · Nature Communications]

Reviewers' comments:

Reviewer #1 (Remarks to the Author):

The authors tried to classify the monolithic glass into two categories, one is with quenched-in nuclei and one is chemically homogeneous with no such nuclei. Using FDSC, they showed nicely the two critical cooling rates for formation of these two types of glasses.

However, I have some concerns and comments as below:

1, Since classification of the glass into two types of monolithic glass is the main goal, the kind of glasses studied should be more than one. The conclusion made may be associated with the particularly alloy used in this study, i.e. easy to have the quenching-in nuclei, just like the Al based alloys [Ref 54-56]. The Zr based glasses do not demonstrate this behavior for example. So I am not so sure if all the glasses can be classified into just two types, as long as the cooling rate is different. It is well known that glasses have many energy states e.g. depending on cooling rates. Classification into two kinds may be too simple.

2, Glasses with nuclei can also be attained by many other ways, e.g. heat treatment. How to classify these? It is not easy to classify the glass into two. It is highly possible, "self-doped glasses" state may not exist for many glasses.

3, They claim that the glass quenched at high cooling rate is homogeneous and at low cooling rate is still amorphous but with quenching-in nuclei, but there is no direct microstructural evidence. The sample may be a "composite" (page 3).

Reviewer #2 (Remarks to the Author):

Metallic glasses are novel materials attracted extensive interests in the past decades because of their unique properties e.g. excellent mechanical properties, physical and chemical performances and also corrosion-resistant property. However, the preparation of materials with amorphous structure is difficult. In particular, whether the structure is the perfect "amorphous" remains a scientific issue. In addition, the control of the cooling or heating processes of metallic glasses is crucial to control the related microstructure and also the subsequent properties. The authors employed the fast differential scanning calorimetry to research the microstructure of Au-based metallic glass, and classified the as-prepared material into two types of amorphous/monolithic glass. The choice of Au-based metallic glass is smart because of its high stability. In particular, the combination of typical metastable material (namely metallic glass in this manuscript) and fast scanning calorimetry will attract extensive attention not only in thermal analysis but also metallic material fields. The involved data are interesting and qualified, and so I deem it is qualified to be published in NC. However, some revisions are advised before its possible acceptance, listed as follows:

1) For the utilized Au₄₉Ag_{5.5}Pd_{2.3}Cu_{26.9}Si_{16.3}, the authors should offer the detailed nominal composition calculated in at.% or wt.%.

2) Flash DSC⁺ instrument was employed, the measurement precision should be given to show its possibility to get the infinitesimal signals of the possible crystallization.

3) The formula sequence is strange, the first is (1) yet the second became (7), and in this case the authors should check this problem.

4) The authors cited Ref. 56 to show that even the TEM and XRD measurements have shown that a BMG appears completely amorphous yet it still possibly contains quenched-in nuclei. There is an advice yet not forced requirement: To characterize the microstructure of two types of amorphous/monolithic glass by high-resolution TEM based on the samples treated by fast scanning calorimetry.

5) In the Supplementary Information section, two questions existed: a) the start formation number is from (2); b) In Fig. 2, two curves isothermally treated at 294°C were applied, yet the curves were different. The authors did not give the reasons why two curves were displayed and why the data were different.

Response to Reviewers

Authors:

We wish to thank the reviewers for acknowledging our work and for recognizing its importance. We also appreciate the reviewers' suggestions, which helped us to improve the paper and to document the discovered effects more clearly. Changes have been made as explained in the following and as identified in the highlighted versions of the revised manuscript and supplementary information.

Reviewers' comments:

Reviewer #1 (Remarks to the Author):

The authors tried to classify the monolithic glass into two categories, one is with quenched-in nuclei and one is chemically homogeneous with no such nuclei. Using FDSC, they showed nicely the two critical cooling rates for formation of these two types of glasses.

However, I have some concerns and comments as below:

I, Since classification of the glass into two types of monolithic glass is the main goal, the kind of glasses studied should be more than one. The conclusion made may be associated with the particularly alloy used in this study, i.e. easy to have the quenching-in nuclei, just like the Al based alloys [Ref 54-56]. The Zr based glasses do not demonstrate this behavior for example. So I am not so sure if all the glasses can be classified into just two types, as long as the cooling rate is different. It is well known that glasses have many energy states e.g. depending on cooling rates. Classification into two kinds may be too simple.

Authors:

We have now also studied a second bulk metallic glass, namely the Pt-based glass Pt_{57.3}Cu_{14.6}Ni_{5.3}P_{22.8}. This glass also shows the various types of SCG, SDG, and CHG. We thus expect this to be a general phenomenon of glasses formed upon cooling. (As mentioned by the reviewer, Al-alloys show pronounced quenched-in nuclei formation upon cooling, meaning that the states of SCG, SDG, and CHG are also expected to exist there.) The measurements of the Pt-based glass have been added to the Supplementary Information: see the section on *Supplementary Figure 2: SDGs and CHGs in a Pt-based glass* (and the corresponding *Supplementary Figure 2*). Corresponding information has also been added to pages 9 and 19 of the manuscript.

Changes to the manuscript:

- Pages 3 and 4, Supplementary Information
- Page 9 and 19, main text

2, Glasses with nuclei can also be attained by many other ways, e.g. heat treatment. How to classify these ? It is not easy to classify the glass into two. It is highly possible, “self-doped glasses” state may not exist for many glasses.

Authors:

In this work we concentrate on glass formation upon linear cooling. The influence of other treatments to form nuclei was not studied. Nevertheless, we expect that classification into CHG and SDG will also hold for other thermal treatments. We have modified the main text accordingly on page 3 (mentioning linear heating and cooling) and page 19 (mentioning other kinds of thermal treatment).

Changes to the manuscript:

- Pages 3 and 19, main text

3, They claim that the glass quenched at high cooling rate is homogeneous and at low cooling rate is still amorphous but with quenching-in nuclei, but there is no direct microstructural evidence. The sample may be a “composite” (page 3).

Authors:

Please note that in this manuscript we distinguish between SCG and the two new amorphous states SDG and CHG. An SCG (semi-crystalline glass) is also called a composite. This was stated in the manuscript, on pages 2 and 3: *“In polymeric materials the terms “amorphous glass” and “semicrystalline glass” are commonly used and distinguished from one another. For metals, a somewhat different terminology is used and the equivalent terms are “monolithic glass” and “metallic glass composite”, respectively.”*

It is also clear from various figures that an SCG strongly differs from an SDG (or a CHG); see Figs. 2, 4, and 5 plus the new Supplementary Figure 2 for the Pt-based glass.

Reviewer #2 (Remarks to the Author):

Metallic glasses are novel materials attracted extensive interests in the past decades because of their unique properties e.g. excellent mechanical properties, physical and chemical performances and also corrosion-resistant property. However, the preparation of materials with amorphous structure is difficult. In particular, whether the structure is the perfect “amorphous” remains a scientific issue. In addition, the control of the cooling or heating processes of metallic glasses is crucial to control the related microstructure and also the subsequent properties. The authors employed the fast differential scanning calorimetry to research the microstructure of Au-based metallic glass, and classified the as-prepared material into two types of amorphous/monolithic glass. The choice of Au-based metallic glass is smart because of its high stability. In particular, the combination of typical metastable material (namely metallic glass in this manuscript) and fast scanning calorimetry will attract extensive attention not only in thermal analysis but also metallic material fields. The involved data are interesting and qualified, and so I deem it is qualified to be published in NC. However, some revisions are advised before its possible acceptance, listed as follows:

1) For the utilized Au₄₉Ag_{5.5}Pd_{2.3}Cu_{26.9}Si_{16.3}, the authors should offer the detailed nominal composition calculated in at.% or wt.%.

Authors:

The composition is given in at.%. The mass loss during alloying was negligible, so that the real composition equaled the nominal one. This information has now been added to the Methods Section (p. 20) of the manuscript.

Changes to the manuscript:

- P. 20, Methods section of the main text

2) Flash DSC2+ instrument was employed, the measurement precision should be given to show its possibility to get the infinitesimal signals of the possible crystallization.

Authors:

We have added details regarding the resolution of the enthalpy measurements to the Supplementary Information. See the section on *Supplementary Methods: Thermal lag*,

temperature calibration, and enthalpy resolution (plus the corresponding Supplementary Figure 5). Corresponding information has also been added to the Methods section in the main text.

Changes to the manuscript:

- Pages 6–8, Supplementary Information
- Page 20, Methods section of the main text

3) The formula sequence is strange, the first is (1) yet the second became (7), and in this case the authors should check this problem.

Authors:

Thank you. We have ordered the formula sequence.

Changes to the manuscript:

- Page 10, main text

4) The authors cited Ref. 56 to show that even the TEM and XRD measurements have shown that a BMG appears completely amorphous yet it still possibly contains quenched-in nuclei. There is an advice yet not forced requirement: To characterize the microstructure of two types of amorphous/monolithic glass by high-resolution TEM based on the samples treated by fast scanning calorimetry.

Authors:

Thank you for this advice. SEM, XRD, and conventional high-resolution TEM can determine metastable microstructures and nano-scale precipitates (see, e.g., Pogatscher et al., *J. Phys.: Condens. Matter.* (2018)). To distinguish between SDG and CHG other techniques such as electron correlation microscopy (see, e.g., Zhang et al., *Nat. Comm.* (2018)) may be needed. However, such measurements were not the focus of this study. We have added a remark on p. 17 in the main text. (In order not to exceed 70 references, we have also deleted refs. 65 and 69 of the original manuscript.)

Changes to the manuscript:

- Page 17, main text

5) In the Supplementary Information section, two questions existed: a) the start formation number is from (2); b) In Fig. 2, two curves isothermally treated at 294°C were applied, yet the curves were different. The authors did not give the reasons why two curves were displayed and why the data were different.

Authors:

Please note that the isothermal curves are influenced by the stochastics of nucleation. Thus, the onset and peak times of crystallization may fluctuate. To illustrate this effect we display two measuring curves for an isothermal hold of 294 °C. As expected, fluctuation for the CHG is greater than for the SDG. To emphasize this, we have added a comment to page 5 of the Supplementary Information.

Changes to the manuscript:

- Page 5, Supplementary Information

REVIEWERS' COMMENTS:

Reviewer #1 (Remarks to the Author):

I have nothing to add.

Reviewer #2 (Remarks to the Author):

I have carefully checked the responses to all the comments from the authors, and I deem these comments are qualified and satisfied. So in this case I agree with the acceptance of this manuscript.